# Multimodal LLM Enhanced Cross-lingual Cross-modal Retrieval

## ABSTRACT

Cross-lingual cross-modal retrieval aims to retrieve visually relevant content based on non-English queries, without relying on human-labeled cross-modal data pairs during training. One popular approach involves utilizing machine translation (MT) to create pseudo-parallel data pairs, establishing correspondence between visual and non-English textual data. However, aligning their representations poses challenges due to the significant semantic gap between vision and text, as well as the lower quality of non-English representations caused by pre-trained encoders and data noise. To overcome these challenges, we propose LECCR, a novel solution that incorporates the multi-modal large language model (MLLM) to improve the alignment between visual and non-English representations. Specifically, we first employ MLLM to generate detailed visual content descriptions and aggregate them into multi-view semantic slots that encapsulate different semantics. Then, we take these semantic slots as internal features and leverage them to interact with the visual features. By doing so, we enhance the semantic information within the visual features, narrowing the semantic gap between modalities and generating local visual semantics for subsequent multi-level matching. Additionally, to further enhance the alignment between visual and non-English features, we introduce softened matching under English guidance. This approach provides more comprehensive and reliable inter-modal correspondences between visual and non-English features. Extensive experiments on two cross-lingual image-text retrieval benchmarks, Multi30K and MSCOCO, as well as two cross-lingual video-text retrieval benchmarks, VATEX and MSR-VTT-CN, demonstrate the effectiveness of our proposed method.

## CCS CONCEPTS

• **Information systems → Multimedia and multimodal retrieval**.

## KEYWORDS

cross-lingual transfer, cross-modal retrieval, Multimodal LLM

**ACM Reference Format:**
Anonymous Author(s). 2024. Multimodal LLM Enhanced Cross-lingual Cross-modal Retrieval . In *Melbourne '32: ACM Symposium on Neural Gaze Detection, June 03–05, 2024, Melbourne, Australia.* ACM, New York, NY, USA, 10 pages. https://doi.org/XXXXXXX.XXXXXXX

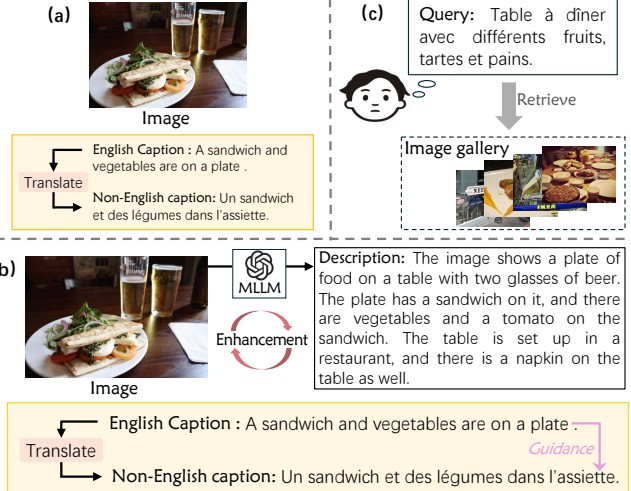

**Figure 1: The comparison between previous methods and our proposed method for CCR. (a) Previous methods typically trained on a collection consisting of images/videos paired with English captions and their corresponding non-English translations. (b) Our approach leverages MLLM to generate detailed visual descriptions and uses them as internal representations to enhance visual representations. Additionally, we utilize English features as guidance to improve alignment between visual and non-English features. (c) During inference, a non-English query is given to retrieve relevant visual content.**

## 1 INTRODUCTION

Cross-lingual cross-modal retrieval (CCR) aims to develop models that can retrieve relevant visual context based on non-English queries, without relying on human-labeled non-English cross-modal data pairs. Compared to traditional cross-modal retrieval, CCR goes beyond the limitations of English and can transfer to other languages. Currently, most research [23, 34–36, 47, 50] in this field resorts MT to generate pseudo-parallel data pairs, as depicted in Figure 1 (a). These studies have achieved remarkable performance by establishing direct correspondences between visual and non-English data.

However, the representation quality of non-English captions tends to exhibit inferior performance compared to English data, which can be attributed to the limitations of pre-trained text encoders when handling non-English languages and the quality of translations. Consequently, aligning visual features with non-English features poses a significant challenge. To overcome this challenge, some methods [23, 27, 47, 50] adopt a single-stream structure (as shown in Figure 2 (a)), utilizing cross-modal fusion modules to capture fine-grained interactions between the modalities. While these methods have demonstrated significant performance improvements, they suffer from increased computational cost and inference time. This is because all possible query-candidate pairs need to be

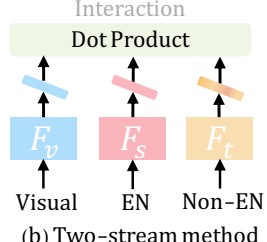
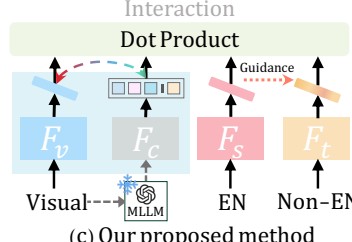

(a) Single-stream method          (b) Two-stream method          (c) Our proposed method

**Figure 2: Comparison of CCR methods. Among them, "$F_v$", "$F_c$", "$F_s$", and "$F_t$" represent the visual encoder, description encoder, English encoder, and non-English encoder, respectively. Our method follows the two-stream structure and incorporates MLLM to enhance the semantics within visual features, helping bridge the gap between modalities. Additionally, to improve the alignment between visual and non-English features, we propose employing English representations as guidance. This can provide more comprehensive and reliable inter-modal correspondence for visual and non-English features.**

processed by the fusion modules to calculate similarity scores. In contrast, alternative approaches [34–36] employ a more efficient dual-stream structure (as shown in Figure 2 (b)), where similarity is calculated using the dot product of their global features. They improve the cross-lingual cross-modal alignment by introducing noise-robust designs. However, these methods extract visual and linguistic features independently without any information fusion, leading to a persistent semantic gap between modalities. Additionally, the image (or video) often conveys richer content than text, as the saying goes, "a picture is worth a thousand words." The captions in existing datasets are typically brief and may only capture partial visual information (see Figure 1 (a)). Therefore, relying solely on global features extracted independently may not fully achieve semantic alignment between visual and non-English features.

Considering the exceptional capabilities demonstrated by MLLMs such as GPT4 [1] and videochat2 [19] in multimodal understanding, we argue that we can generate complementary contextual descriptions using MLLM to enhance the semantics within visual features (as depicted in Figure 1 (b) and Figure 2 (c)). This approach enables us to bridge the semantic gap between modalities more effectively. Similarly, other works [25, 38, 40, 41, 44, 45] have also incorporated MLLMs to enhance existing vision-language models. However, these works simply utilize the [CLS] token or all descriptions to interact with visual features, potentially resulting in incomplete context information or high computational costs. Therefore, it is crucial to further investigate *how to effectively utilize these visual descriptions to enhance visual representations, thereby improving the alignment between visual and non-English features.*

To address the aforementioned challenges, we propose LECCR (Multimodal LLM Enhanced Cross-lingual Cross-modal Retrieval), a novel two-stream solution. Specifically, given the descriptions generated by MLLM, we first aggregate these descriptions into the multi-view semantic slots, and incorporate the regularization loss to force these semantic slots to focus on the different semantics present in descriptions (*e.g.,* different objects in the image). Subsequently, we introduce a multi-view visual-semantic interaction module to interact these semantic slots with the visual features. This module not only enhances the semantics within visual features but also generates local visual context semantics, enabling multi-level cross-lingual cross-modal matching and effectively narrowing the semantic gap between modalities. Finally, we use English features

as guidance to establish a more comprehensive correspondence between visual and non-English features.

The proposed method, LECCR, is evaluated and compared with previous work on two text-image retrieval benchmarks, Multi30K and MSCOCO, as well as two text-video retrieval benchmarks, VATEX and MSR-VTT-CN. Our method consistently outperforms previous approaches in the majority of evaluation settings, highlighting its effectiveness in CCR task. Our contributions can be summarized as follows: (1) We propose a new two-stream CCR solution, LECCR, which incorporates MLLM to improve the alignment between visual and non-English features. (2) To bridge the semantic gap between modalities, we utilize detailed visual descriptions generated by MLLM and aggregate them into multi-view semantic slots to enhance visual features. We then introduce multi-level matching and softened matching under English guidance to improve the alignment between visual and non-English features. (3) We conduct extensive experiments on four image-text and video-text cross-modal retrieval benchmarks across different languages, demonstrating the effectiveness and potential of our method.

## 2  RELATED WORK

### 2.1  Cross-lingual Cross-modal Retrieval (CCR)

CCR has been attracting growing interest among researchers. Compared with the traditional cross-modal retrieval [3, 7, 11, 14, 16–18, 20, 24, 37, 49], this approach offers a highly efficient and cost-effective solution for non-English-based retrieval, reducing the reliance on human-labeled data. Existing methods can be grouped into two categories in terms of the model architecture. The first approach [23, 27, 47, 50] adopts a single-stream structure that incorporates cross-lingual cross-modal fusion modules to model both image regions and multilingual text word representations in a unified semantic space, capturing the fine-grained relationship between them. For instance, Ni et al. [27] use a code-switch strategy and masked modeling loss to model the interaction between vision and multiple languages. However, this approach includes an extra cross-modal fusion module, which may lead to computation overhead and slower inference speeds. Therefore, it may not be practical for large-scale CCR tasks in real-world settings.

The second approach [13, 15, 34–36] involves a two-stream structure, where each stream is dedicated to modeling either the vision

or language inputs. For instance, Jain et al. [15] use scalable dual encoder models trained with contrastive losses to learn encoders for both language and images, combining image-text matching and text-text matching tasks. This approach maps the global features of different modalities into a common semantic space to learn cross-modal alignment. However, despite its efficiency, this approach still encounters challenges due to the absence of explicit cross-modal interaction. Additionally, existing dataset captions are typically brief and may only capture partial visual information. The information imbalance between two modalities further exacerbating the difficulty in aligning their features. In this paper, we adopt the two-stream structure and incorporate the MLLM to provide additional contextual semantics for visual features. This can help narrow the semantic gap between modalities and further improve alignment between visual and non-English features.

## 2.2 LLM-enhanced Vision-Language Models

With exceptional language understanding capabilities, LLMs [2, 30, 31, 31] have exhibited remarkable capabilities in a variety of tasks, such as image captioning, and visual question answering. Recently, a new line of work has emerged [25, 28, 38, 40, 41, 44, 45] that incorporates them to enhance the vision-language models (VLMs). For example, in the classification task, menon *et al.,* [25] and pratt *et al.,* [28] leverage the LLMs to generate the textual descriptors for each category, comparing images to these descriptors rather than estimating the similarity of images directly with category names. Similarly, Yang *et al.,* [41] use GPT-3 along with text descriptions of images for the Visual Question Answering (VQA) task. In this work, we extend the descriptive LLMs to generate visual context descriptions in a CCR task. Another relevant work in cross-modal retrieval, Wu et al. [38], employs LLMs to generate auxiliary captions to enhance text-video matching. However, their approach primarily utilizes auxiliary captions for data augmentation and simply interacts with visual features using [CLS] embeddings. In contrast, our objective is to leverage rich descriptions to offer semantic context for visual features. Moreover, we introduce multi-view semantic slots to comprehensively represent the description content, thereby providing contextual semantic information for visual features.

## 3 METHODS

Figure 3 shows an overview of our approach. In what follows, we will briefly describe the CCR definitions and our Baseline method (in Section 3.1). Then, we present our method LECCR, including multi-view semantic slots generation (in Section 3.2), multi-view visual-semantic interaction (in Section 3.3), multi-level matching (in Section 3.4), and softened matching under English guidance (in Section 3.5), respectively.

## 3.1 Preliminary

The objective of the CCR task is to retrieve the relevant visual content (*i.e.,* image or video ) using the non-English query (see Figure 1 (c)), while solely relying on annotated paired visual-English sample pairs during training. Following previous studies [34, 36, 47], we employ the MT to generate the translated captions based on the English captions. This enables us to construct a dataset consisting of triplet sample pairs $\mathcal{D} = (V, S, T)$, where $V$, $S$, and $T$ represent

the visual items (*e.g.,* images or videos), English captions, and non-English captions, respectively. Then, we take them as the input, and use the vision encoder $\mathcal{F}_v$, English encoder $\mathcal{F}_s$, and non-English encoder $\mathcal{F}_t$ to extract the corresponding sequential representations $\mathbf{Z}_v \in \mathbb{R}^{N_v \times d_v}$, $\mathbf{Z}_s \in \mathbb{R}^{N_s \times d_s}$, and $\mathbf{Z}_t \in \mathbb{R}^{N_t \times d_t}$, where $N_{x \in \{v,s,t\}}$ denotes the length of each sequence, and $d_{x \in \{v,s,t\}}$ denotes the channel dimension. Finally, we project them into a multi-lingual multi-modal common space. This process can be represented as:

$$\mathbf{Z}_v = \mathcal{F}_v(V), \mathbf{Z}_s = \mathcal{F}_s(S), \mathbf{Z}_t = \mathcal{F}_t(T), \tag{1}$$

$$\mathbf{h}_v = \phi_v(\mathbf{Z}_s^{cls}), \mathbf{h}_s = \phi_s(\mathbf{Z}_s^{cls}), \mathbf{h}_t = \phi_t(\mathbf{Z}_t^{cls}) \tag{2}$$

where $\phi(\cdot)$ denotes the linear projection function used to project the features into a common space, $\mathbf{Z}_x^{cls}$ is the [CLS] features, and $\mathbf{h}_x \in \mathbb{R}^d$ denotes the corresponding latent features in the common space, where $x \in \{v, s, t\}$.

Following this, we introduce the contrastive loss to pull the paired samples closer to each other and push the non-paired samples away from each other. It can be defined as:

$$\mathcal{L}_{vs} = \mathcal{L}_{ctra}(\mathbf{h}_v, \mathbf{h}_s), \mathcal{L}_{ts} = \mathcal{L}_{ctra}(\mathbf{h}_t, \mathbf{h}_s), \mathcal{L}_{vt} = \mathcal{L}_{ctra}(\mathbf{h}_v, \mathbf{h}_t) \tag{3}$$

Among them, the contrastive loss $\mathcal{L}_{ctra}$ can be formulated as:

$$\mathcal{L}_{ctra}(\mathbf{a}, \mathbf{b}) = -\frac{1}{2} \times \frac{1}{B} \sum_{i=1}^{B} [log \frac{exp(S_g(\mathbf{a}^i, \mathbf{b}^i)/\tau)}{\sum_{j=1}^{B} exp(S_g(\mathbf{a}^i, \mathbf{b}^j)/\tau)}$$
$$+ log \frac{exp(S_g(\mathbf{a}^i, \mathbf{b}^i)/\tau)}{\sum_{j=1}^{B} exp(S_g(\mathbf{a}^j, \mathbf{b}^i)/\tau)}] \tag{4}$$

where $B$ represents the mini-batch size, $\tau$ represents the temperature coefficient, and $S_g(\mathbf{a}^i, \mathbf{b}^j) = \frac{\mathbf{a}^{i^T} \mathbf{b}^j}{||\mathbf{a}^i|| \cdot ||\mathbf{b}^j||}$ represents the similarity function to calculate the similarity between the $i$-th feature vector $\mathbf{a}$ and the $j$-th feature vector $\mathbf{b}$. The final objective can be calculated as: $\mathcal{L}_{base} = \mathcal{L}_{vs} + \mathcal{L}_{ts} + \mathcal{L}_{vt}$.

During the inference, we calculate the similarities $S_g(\mathbf{h}_v, \mathbf{h}_t)$ between the visual features $\mathbf{h}_v$ and the non-English features $\mathbf{h}_t$ to perform retrieval. Note that we take the above process as our Baseline method.

## 3.2 Multi-view Semantic Slots Generation

In this section, we aim to aggregate visual descriptions generated by MLLM into the multi-view semantic slots, enabling us to capture diverse semantics within them. This is substantially different from existing methods that only utilize the [CLS] token or all representations, which may not fully leverage the contextual information in the descriptions. Specifically, given the images (or videos), we feed them into the MLLM with the prompt, such as *"### Please describe the contents of this image in detail."* to generate the rich visual descriptions $C$ ( as illustrated in Figure 4). It is worth noting that our visual descriptions are generated in English, and no additional translations are introduced, avoiding any extra loss in quality. Then, we extract description embeddings $\mathbf{Z}_c \in \mathbb{R}^{N_c \times d_c}$ using the description encoder $\mathcal{F}_c$ and employ the $N_q$ learnable queries that are randomly initialized to aggregate them. This aggregation process enables us to generate multi-view semantic slots

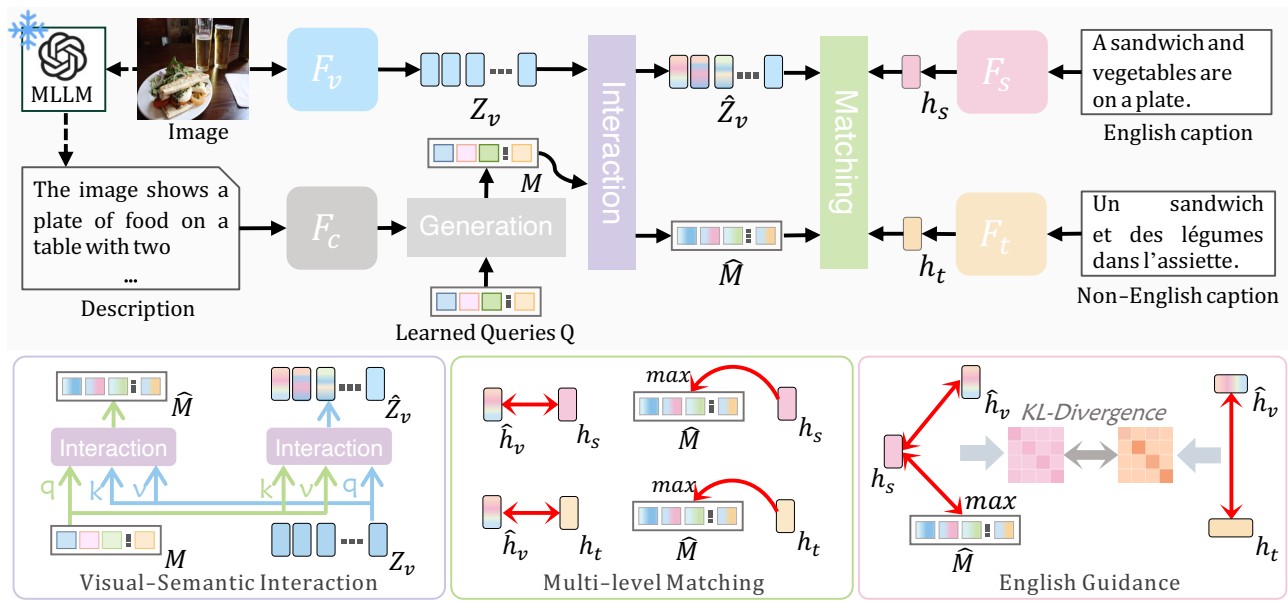

**Figure 3: Overview of the proposed LECCR framework. We utilize the multi-modal large language model (MLLM) to generate detailed visual descriptions, which are then employed as internal features to enhance the visual representations. Additionally, we introduce multi-level matching and softened matching under English guidance to improve the alignment between visual and non-English representations.**

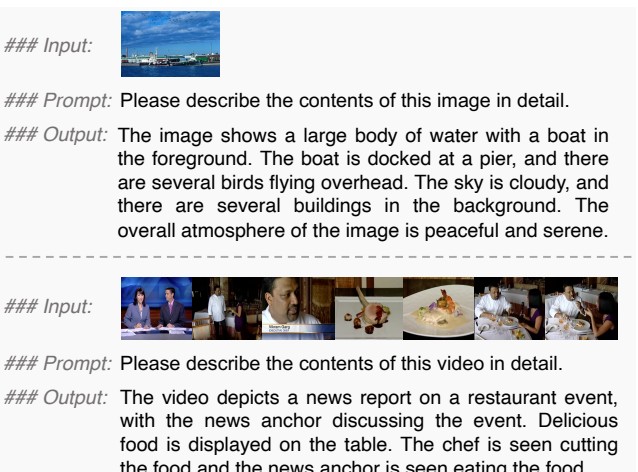

**Figure 4: The example of the visual description generated using MLLM.**

$\mathbf{M} \in \mathbb{R}^{N_q \times d}$, where each view encapsulates distinct semantics of input descriptions. Mathematically, this can be represented as:

$$\mathbf{Z}_c = \phi_c(\mathcal{F}_c(C)) \tag{5}$$

$$\bar{\mathbf{Q}} = MHCA(\mathbf{Q}, \mathbf{Z}_c, \mathbf{Z}_c) \tag{6}$$

$$\mathbf{M} = LN(\phi_q(\bar{\mathbf{Q}})) + \bar{\mathbf{Q}} \tag{7}$$

where $MHCA(,,)$ represents the multi-head cross-attention module, $\phi_c(\cdot)$ and $\phi_q(\cdot)$ represent the linear projection functions, and $LN(\cdot)$ represents the layer norm.

Additionally, to prevent the generated multi-view semantic slots from overly focusing on the same semantic features, we introduce a multi-view regularization loss to encourage diversity among the slots. This objective can be formulated as:

$$\mathcal{L}_{reg} = -\frac{1}{B} \times \frac{1}{N_q} \sum_{i=1}^{B} \sum_{j=1}^{N_q} log P^{i,j} \tag{8}$$

where $p^{ij} = \frac{exp(\mathbf{m}^{ij T} \mathbf{m}^{ij})}{\sum_{k=1}^{N_q} exp(\mathbf{m}^{ij T} \mathbf{m}^{ik})}$ represents the similarity distribution between views, and $\mathbf{m}^{ij}$ denotes the $j$-view semantic slot of the $i - th$ sample.

### 3.3 Multi-view Visual-Semantic Interaction

We propose a vision-semantic interaction module that uses the above multi-view semantic slots as the internal representation to bridge the semantic gap between modalities. This module serves two primary purposes: (1) semantic slots to vision (C2V), which provides additional contextual semantic information to enhance the semantics within visual features; (2) vision to semantic slots (V2C), enabling the multi-view semantic slots to capture their corresponding visual information and generate local contextual visual semantics. Specifically, we take the visual representations $\mathbf{Z}_v$ and the multi-view semantic slots $M$ as inputs to the dual attention block. This module enables us to generate two outputs: semantic-enhanced visual features $\hat{\mathbf{Z}}_v$ and local contextual visual semantics $\hat{\mathbf{M}}$. Here, we present two alternative options for the interaction as below:

(1) Dual cross-attention: we utilize two cross-attention blocks, where the visual features $\mathbf{Z}_v$ and multi-view semantic slots $\mathbf{M}$ are

used as queries, respectively. These blocks then aggregate information from the features of each other. It can be formally defined as:

$$\bar{\mathbf{Z}}_v = MHCA(\mathbf{Z}_v, \mathbf{M}, \mathbf{M}), \quad \bar{\mathbf{M}} = MHCA(\mathbf{M}, Z_v, Z_v) \quad (9)$$

$$\hat{\mathbf{Z}}_v = LN(\phi_z(\bar{\mathbf{Z}}_v)) + \bar{\mathbf{Z}}_v, \quad \hat{\mathbf{M}} = LN(\phi_m(\bar{\mathbf{M}})) + \bar{\mathbf{M}} \quad (10)$$

where $\phi_z(\cdot)$ and $\phi_m(\cdot)$ represent the linear projection functions.

(2) Co-attention: we concatenate the visual features $\mathbf{Z}^v$ and semantic slots $\mathbf{M}$ together and process them through a self-attention block:

$$\bar{\mathbf{Z}}_v, \bar{\mathbf{M}} = MHSA([\mathbf{Z}_v; \mathbf{M}], [\mathbf{Z}_v; \mathbf{M}], [\mathbf{Z}_v; \mathbf{M}]) \quad (11)$$

$$\hat{\mathbf{Z}}_v = LN(\phi_z(\bar{\mathbf{Z}}_v)) + \bar{\mathbf{Z}}_v, \quad \hat{\mathbf{M}} = LN(\phi_m(\bar{\mathbf{M}})) + \bar{\mathbf{M}} \quad (12)$$

where $MHSA(,,)$ represents the multi-head self-attention, and $[;]$ represents the concatenate operation along the length dimension of the sequence.

## 3.4 Multi-level Matching

After the vision-semantic interaction, the multi-view semantic slots aggregate the corresponding local contextual visual features, which can be considered as the local features. Next, we introduce multi-level matching, including caption-slots matching (local level) and caption-vision matching (global level), to facilitate cross-lingual cross-modal alignment.

**Caption-slots matching.** Considering that semantic slots from each view may be associated with different aspects of the visual content, there could be significant semantic disparities between the different views. Hence, we align the caption with the slot whose semantics are most closely related, instead of imposing strict alignment constraints on all view semantic slots. To achieve this, the similarity scores between the caption and the semantic slots are computed as follows:

$$S_l(\mathbf{h}_x^i, \hat{\mathbf{m}}^j) = \max_{0 \le k \le N_q} \frac{(\hat{\mathbf{m}}^{jk})^T \mathbf{h}_x^i}{||\hat{\mathbf{m}}^{jk}|| \cdot ||\mathbf{h}_x^i||}, x \in \{s, t\} \quad (13)$$

where $\mathcal{S}_l(\mathbf{h}_x^i, \hat{\mathbf{m}}^j)$ represents the similarity score between the $i$-th caption and the $j$-th multi-view semantic slots. Then, the caption-slots matching objective $\mathcal{L}_c$ can be formulated as:

$$\mathcal{L}_{sc} = -\frac{1}{B} \sum_{i=1}^{B} \log \frac{exp(S_l(\mathbf{h}_s^i, \hat{\mathbf{m}}^i)/\tau)}{\sum_{j=1}^{B} exp(S_l(\mathbf{h}_s^i, \hat{\mathbf{m}}^j)/\tau)} \quad (14)$$

$$\mathcal{L}_{tc} = -\frac{1}{B} \sum_{i=1}^{B} \log \frac{exp(\mathcal{S}_l(\mathbf{h}_t^i, \hat{\mathbf{m}}^i)/\tau)}{\sum_{j=1}^{B} exp(\mathcal{S}_l(\mathbf{h}_t^i, \hat{\mathbf{m}}^j)/\tau)} \quad (15)$$

$$\mathcal{L}_c = \frac{1}{2}(\mathcal{L}_{sc} + \mathcal{L}_{tc}) \quad (16)$$

**Caption-vision matching.** Similar to Equation 3, in this case, we utilize the semantic-enhanced visual features $\hat{\mathbf{Z}}_v$ to align with the caption features, represented as:

$$\mathcal{L}_{vs} = \mathcal{L}_{ctra}(\hat{\mathbf{h}}_v, \mathbf{h}_s), \ \mathcal{L}_{vt} = \mathcal{L}_{ctra}(\hat{\mathbf{h}}_v, \mathbf{h}_t) \quad (17)$$

$$\mathcal{L}_v = \frac{1}{2}(\mathcal{L}_{vs} + \mathcal{L}_{vt}) \quad (18)$$

where $\hat{\mathbf{h}}_v = \phi_v(\hat{\mathbf{Z}}_v^{cls})$ represents the global visual features.

## 3.5 Softened matching under English Guidance

The ground-truth labels used in Equation 3 are hard one-hot labels, which assume no correlation between unpaired samples. This approach assigns equal weights to all negative samples, disregarding the potentially valuable inter-modal relationships. This makes aligning visual and non-English features more challenging. To address this issue, we propose utilizing English features as guidance for non-English ones. Our goal is to use the visual-English similarity as a softened target to guide the alignment between visual and non-English features. This softened target helps establish comprehensive relationships between modalities. To capture the relationships between modalities more effectively, we calculate vision-English similarity at multiple levels, including both local and global levels. We then integrate these similarities to generate softened targets that direct the non-English features. This process can be represented mathematically as follows:

$$S_{soft}(\mathbf{h}_s^i, \hat{\mathbf{h}}_v^j, \mathbf{m}^j) = \alpha \cdot S_g(\mathbf{h}_s^i, \hat{\mathbf{h}}_v^j) + (1 - \alpha) \cdot S_l(\mathbf{h}_s^i, \mathbf{m}^j) \quad (19)$$

$$y^{ij} = softmax(S_{soft}(\mathbf{h}_s^i, \hat{\mathbf{h}}_v^j, \mathbf{m}^j)/\tau) \quad (20)$$

where $Y \in \mathbb{R}^{B \times B}$ represents the softened targets, and $\alpha$ represents the weight parameter. Following, we use the KL-Divergence to supervise the visual-non-English correspondence:

$$\bar{y}^{ij} = softmax(S_g(\mathbf{h}_t^i, \hat{\mathbf{h}}_v^j)/\tau) \quad (21)$$

$$\mathcal{L}_{rkt} = \frac{1}{B} \sum_{i=1}^{B} KL(y^i \ || \ \bar{y}^i) \quad (22)$$

Finally, the matching between visual and non-English features in Equation 17 can be modified as:

$$\hat{\mathcal{L}}_{vt} = \lambda \cdot \mathcal{L}_{vt} + (1 - \lambda) \cdot \mathcal{L}_{rkt} \quad (23)$$

where $\lambda$ represents the weight parameter.

## 3.6 Training and Inference

**Training.** The final training objective can be formulated as:

$$\mathcal{L} = \mathcal{L}_{ts} + \mathcal{L}_v + \mu_1 \mathcal{L}_c + \mu_2 \mathcal{L}_{reg} \quad (24)$$

where $\mu_1$ and $\mu_2$ represent the loss weights. In the experiments, we set them to 0.1 and 0.01, respectively.

**Inference.** After training the model, given a sentence query in non-English, we sort candidate videos/images in descending order based on their similarity scores with the query. Specifically, we first compute the similarity between visual features $\hat{h}^v$ and query features $h^t$, as well as between semantic slots $\hat{M}$ and query features $h^t$. Then, we combine the two scores to obtain the final similarity score. This can be formulated as:

$$S_{final} = \beta \cdot S_g(\mathbf{h}_t, \hat{\mathbf{h}}_v) + (1 - \beta) \cdot S_l(\mathbf{h}_t, \hat{\mathbf{M}}) \quad (25)$$

where $\beta$ denotes the weight parameter, which we set to 0.8 in our experiments.

# 4 EXPERIMENT

## 4.1 Experimental Settings

**Datasets.** We conduct experiments on two public multilingual image-text retrieval datasets (Multi30K [10]) and MSCOCO[6]), as well as two video-text retrieval datasets, VATEX [32] and MSR-VTT-CN [34]. Notably, we only use the annotated vision-English data pair in the training process, while using the non-English query (human labeled) to evaluate. Following previous work [34], the MT we adopt the Google translator.

- **Multi30K** [10]: This dataset consists of 31,000 images and is a multi-lingual version of Flickr30K [43]. It involves four languages, *i.e.,* English(en), German(de), French(fr), and Czech(cs). For the dataset partition, we split the data into train/dev/test sets in a 29000/1000/1000, similar to [43].
- **MSCOCO** [6]: This dataset consists of 123,287 images, and each image has 5 captions. We translate the training set from English into Chinese(zh) and Japanese(ja) by resorting to MT, and using the test sets from the [21] and [42], respectively. We follow the data split as in [50].
- **VATEX** [32]: VATEX is an extensive video-text retrieval dataset that provides bilingual captions for over 41,250 videos. Each video is paired with 10 English sentences and 10 Chinese sentences that describe its contents in detail. Similar to the approach taken in [5, 9], we train our models on 25,991 video clips, while we reserve 1,500 clips each for validation and testing.
- **MSR-VTT-CN** [34]: MSR-VTT-CN is the multilingual version of the MSR-VTT, which covers English and Chinese. We follow the partition of [46] containing 9,000 and 1,000 for the training and testing, respectively.

**Evaluation metrics.** For video-text retrieval, we follow the previous works [22, 33], and use rank-based metrics, namely $R@K$ ($K = 1, 5, 10$), and the sum of all Recalls (SumR) to evaluate the performance. $R@K$ is the fraction of queries that correctly retrieve desired items in the top $K$ of ranking list. Higher $R@K$ and mAP mean better performance. For image-text retrieval, we only report the SumR.

**Implementation details** Following [34], we use the CLIP [29] to extract the image representations, and use mBERT [8] to extract the text representations. For video features, on MSR-VTT-CN, we use the frame-level features ResNet-152 [12] and concatenate frame-level features ResNeXt-101 [39][26] to obtain a combined 4,096-dimensional feature. On VATEX, we adopt the I3D [4] video feature which is officially provided. For all experiments, we train the modal for 40 epochs with a cosine decay scheduler and an initial learning rate of $1 \times 10^{-5}$. We use the videochat2 to generate the visual descriptions for images and videos. Besides, all text encoders are parameters shared.

## 4.2 Evaluation on Cross-lingual Image-Text Retrieval

We conducted a comprehensive evaluation of our LECCR method compared to state-of-the-art approaches on two widely used image-text datasets, Multi30K and MSCOCO. It is worth noting that M³P, UC², UMVLP, CCLM, MURAL, and MLA were pre-trained on large-scale vision-language datasets, while NRCCR, DCOT, CL2CM, and

**Table 1: Cross-lingual image-text retrieval results on Multi30K and MSCOCO. Following previous work, we use the sumR as the metric. \*: Models pre-trained on large-scale datasets, e.g., CC3M and its MT version. †: Model uses the same initialization parameters with CCLM. "XLMR-L/-B" denote the XLMR-Large/-Base. The single-stream method is usually a one-to-one matching Siamese architecture, so its inference efficiency is lower than that of the two-stream method. During inference, LECCR† is about 10x faster than CCLM.**

| Method | Backbone | Multi30K | | | MSCOCO | |
| --- | --- | --- | --- | --- | --- | --- |
| | | en2de | en2fr | en2cs | en2zh | en2ja |
| **Single-Stream:** | | | | | | |
| M³P [27]* | XLMR-L | 351.0 | 276.0 | 220.8 | 332.8 | 336.0 |
| UC² [50]* | XLMR-B | 449.4 | 444.0 | 407.4 | 492.0 | 430.2 |
| UMVLP [23]* | XLMR-L | 506.4 | 516.6 | 463.2 | 499.8 | 438.6 |
| CCLM [47]* | XLMR-L | **540.0** | **545.4** | **536.4** | **546.0** | **532.8** |
| **Two-Stream:** | | | | | | |
| MURAL [15]* | XLMR-L | 456.0 | 454.2 | 409.2 | - | 435.0 |
| MLA [48]* | CLIP | 495.6 | 510.0 | 457.2 | - | 482.4 |
| NRCCR [34] | mBERT | 480.6 | 482.1 | 467.1 | 512.4 | 507.0 |
| DCOT [36] | mBERT | 494.9 | 495.3 | 481.8 | 521.5 | 515.3 |
| CL2CM [35] | mBERT | 498.0 | 498.6 | 485.3 | 522.0 | 515.9 |
| CL2CM [35]† | XLMR-L | 530.4 | 534.1 | 526.3 | 544.3 | 546.2 |
| LECCR | mBERT | 505.2 | 507.8 | 494.3 | 535.7 | 532.8 |
| LECCR† | XLMR-L | **535.6** | **537.0** | **529.8** | **548.1** | **547.7** |

**Table 2: Cross-lingual video-text retrieval results on VATEX (en2zh). \*: Model pre-trained on a large-scale dataset Multi-HowTo100M[13].**

| Method | T2V | | | V2T | | | SumR |
| --- | --- | --- | --- | --- | --- | --- | --- |
| | R@1 | R@5 | R@10 | R@1 | R@5 | R@10 | |
| MMP w/o pre-train[13] | 23.9 | 55.1 | 67.8 | - | - | - | - |
| MMP [13]* | 29.7 | 63.2 | 75.5 | - | - | - | - |
| NRCCR [34] | 30.4 | 65.0 | 75.1 | 40.6 | 72.7 | 80.9 | 364.8 |
| DCOT [36] | 31.4 | 66.3 | 76.8 | 46.0 | 76.3 | 84.8 | 381.8 |
| CL2CM [35] | 32.1 | 66.7 | 77.3 | 48.2 | 77.1 | 85.5 | 386.9 |
| LECCR | **32.7** | **67.9** | **78.8** | **49.0** | **78.8** | **87.2** | **394.4** |

our method do not require additional pre-training data. In Table 1, we can observe that our LECCR achieves better performance than all two-stream methods. Specifically, compared to the baseline method CL2CM, our approach demonstrates improvements of 1.4%, 1.8%, 1.9%, 2.6%, and 3.4% on all languages in terms of sumR. Unlike the single-stream models that incorporate cross-modal fusion modules to capture detailed interactions between image regions and text words, our LECCR incorporates LLM to provide additional semantic context information for visual features. This can narrow the gap between modalities and improve the alignment between visual and non-English representations. Note that our method is xx faster than CCLM in inference times. Moreover, when equipped with a stronger backbone pre-trained on large-scale datasets, our LECCR† achieves comparable performance to the single-stream method CCLM.

**Table 3: Cross-lingual video-text retrieval results on MSR-VTT-CN (en2zh).**

| Method | T2V | | | V2T | | | SumR |
|--------|-----|-----|------|-----|-----|------|------|
| | R@1 | R@5 | R@10 | R@1 | R@5 | R@10 | |
| NRCCR [34] | 29.6 | 55.8 | 67.4 | **31.3** | 56.0 | 67.2 | 307.3 |
| LECCR | **29.8** | **56.5** | **69.1** | 29.5 | **59.1** | **69.1** | **312.5** |

**Table 4: Ablation study of our proposed component. "MVSS", "MM", and "SMEG" represent the multi-view semantic slots generation and visual-semantic interaction, multi-level matching, and softened matching under English guidance, respectively. The "+" symbol indicates that the module is gradually added based on the line above it.**

| Method | en2de | en2fr | en2cs |
|--------|-------|-------|-------|
| Baseline | 484.6 | 491.4 | 482.1 |
| + MVSS | 498.0 | 499.2 | 487.9 |
| + MM | 501.1 | 503.4 | 490.2 |
| + SMEG | **505.2** | **507.8** | **494.3** |

**Table 5: Ablation study of multi-view visual-semantic interaction.**

| Method | en2de | en2fr | en2cs |
|--------|-------|-------|-------|
| **Interaction manner:** | | | |
| Co-attention | 504.4 | 505.1 | 494.0 |
| Dual cross-attention | **505.2** | **507.8** | **494.3** |
| **Interaction direction:** | | | |
| Single interaction V2C | 501.4 | 502.4 | 489.7 |
| Single interaction C2V | 503.4 | 505.0 | 491.9 |
| Dual interaction | **505.2** | **507.8** | **494.3** |

## 4.3 Evaluation on Cross-lingual Video-Text Retrieval

The experimental results on the VATEX and MSR-VTT-CN datasets are reported in Table 2 and Table 3. We can see that our method LECCR outperforms all baseline methods using a two-stream structure. This further demonstrates the effectiveness of our approach in incorporating LLM to provide additional contextual semantic information. Additionally, even when compared to the MMP trained on large-scale multi-lingual multi-modal datasets, our method still achieves superior results.

## 4.4 Ablation Studies

In this section, we provided detailed ablation studies on Multi30K to verify the effectiveness of each part of our design.

**Analysis of each component.** To validate the effectiveness of the components within our model, we conduct ablation experiments on them. As shown in Table 4, we observed consistent improvements in performance as the components were incrementally added.

**Table 6: Ablation study of the soft matching under English guidance.**

| Method | en2de | en2fr | en2cs |
|--------|-------|-------|-------|
| Baseline | 484.6 | 491.4 | 482.1 |
| w/o guidance | 501.1 | 503.4 | 490.2 |
| Guidance with $S_g$ | 503.2 | 504.7 | 492.0 |
| Guidance with $S_l$ | 503.5 | 505.2 | 492.1 |
| Guidance with $S_g$ and $S_l$ | **505.2** | **507.8** | **494.3** |

Specifically, the introduction of multi-view semantic slots and their corresponding interaction resulted in performance increases of 2.8%, 1.6%, and 1.2%, respectively. This clearly demonstrates that the complementary context information provided by MLLM helps bridge the gap between modalities. Furthermore, when we incorporated multi-level matching and English guidance, we observed a significant performance gain.

**Analysis of multi-view visual-semantic interaction.** In Table 5, we investigate various interaction modules mentioned in Section 3.3. Both modules employ a transformer structure and utilize the multi-head self-attention mechanism to fuse visual and semantic information. The results indicate that the dual cross-attention module achieves superior performance, and we use it in the following experiments. Furthermore, we also conduct the experiment to analyze the interaction direction. The results reveal that dual interaction leads to significant performance improvements. This is because dual interaction not only enhances the semantic information within visual features, but also generates local visual semantics that can be utilized for multi-level matching. In summary, the results demonstrate that the multi-view visual-interaction module can lead to better visual representations and improved alignment between visual and non-English features.

**Analysis of soft matching under English guidance.** In Table 6, we conduct the experiment to investigate the effectiveness of soft matching and analyze the impact of the source of the softened targets. Notably, we observed a substantial performance drop when soft matching was removed. This suggests that the rich relations between samples are important to improve the alignment between visual and non-English features. Moreover, the semantic slots after interaction integrate local visual contextual semantics, enabling the capture of more detailed local visual information and helping to provide the better guidance. The results demonstrate that the combination of $S_m$ with $S_g$ further improves the performance.

**Analysis of the description representation.** We conducted a comparison between our proposed multi-view semantic slots and other methods for extracting description representations, namely CLS, Mean, and All. As shown in Table 7, our method consistently outperforms the others. Using the [CLS] token and mean pooling methods both extract global information from descriptions, but this may lead to a loss of rich contextual semantics. Consequently, it becomes challenging to provide comprehensive semantic information for visual features. On the other hand, directly employing all description representations may result in information redundancy and make it difficult to extract useful and diverse key information.

**Table 7: Ablation study of the extraction of MLLM-generated descriptions. "CLS" denotes using the [CLS] token of description representations; "mean" denotes using the mean pooling applied to the description representations; "All" denotes using the all description representations; and "Multi-view" denotes our proposed multi-view semantic slots.**

| Method | en2de | en2fr | en2cs |
|---|---|---|---|
| CLS | 501.1 | 502.3 | 488.8 |
| Mean | 501.8 | 503.0 | 489.0 |
| All | 502.7 | 504.9 | 491.3 |
| Multi-view | **505.2** | **507.8** | **494.3** |

**Table 8: Ablation study on the encoding manner of MLLM-generated descriptions. "Share" denotes the encoder that is a parameter shared with the caption encoder.**

| Method | en2de | en2fr | en2cs |
|---|---|---|---|
| Frozen | 492.1 | 495.0 | 484.9 |
| Finetune | 501.5 | 503.1 | 489.9 |
| Share | **505.2** | **507.8** | **494.3** |

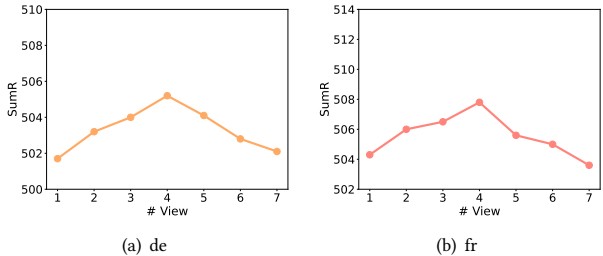

(a) de       (b) fr

**Figure 5: The performance of different numbers (#view) of semantic slots.**

In contrast, our proposed multi-view semantic slots effectively aggregate diverse semantics within descriptions. This enables us to capture rich contextual semantics of visual representations in the subsequent interaction module.

**Analysis of the description encoding.** In Table 8, we perform an ablation study to investigate different approaches to encoding the description representation. The results indicate that utilizing a frozen text encoder to extract the description representation leads to inferior results. In contrast, we find that the best performance is achieved when using shared parameters with the caption encoder. We hypothesize that this is due to the significant distributional differences between description and caption features when using a frozen text encoder. Therefore, we adopt the shared text encoder for description representation encoding, and it does not incur any additional computational costs.

**Analysis of the number of semantic slots.** In Figure 5, we investigate the impact of the number of semantic slots on performance. As shown, performance is best when 4 views are used. When

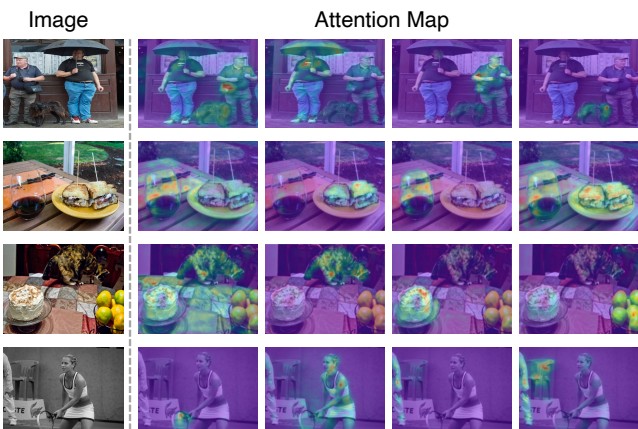

Image       Attention Map

**Figure 6: The visualization of multi-view semantic slots in multi-view semantic interaction module (#view = 4). Each semantic slot can distinctly focus on local semantics within the images.**

more views are added, performance slightly drops. We hypothesize that this is because too many views may lead to information redundancy.

### 4.5 Visualization of Multi-view Semantic Slots.

In Figure 6, attention maps from the last attention layer in the multi-view visual-semantic interaction module (V2C) are visualized. These attentions are designed to capture visual information corresponding to the semantic slots. Through this interaction module, we obtain diverse local visual semantics that specifically focus on different objects or contexts. These semantics are then utilized in subsequent multi-level alignment objectives to improve the alignment between visual and non-English features. As shown, each slot focuses on distinct objects in the images, showcasing the diverse semantics captured by the multi-view semantic slots. For instance, in the third line, the semantic slots highlight the "cat", "oranges", and "cake" respectively in the image. The visualization results provide further evidence of the effectiveness of the interaction module in capturing and integrating related visual information.

## 5 SUMMARY AND CONCLUSIONS

In this paper, we present LECCR, a novel two-stream solution for the CCR task. Our approach aims to bridge the gap between modalities by incorporating the multi-modal large language model (MLLM) to generate detailed visual descriptions. These descriptions serve to provide additional contextual semantics for the visual representations. Additionally, considering the lower quality of non-English representations, we utilize English representations as guidance to improve the alignment between visual and non-English features by providing comprehensive inter-modal relationships. Extensive experiments demonstrate that LECCR significantly improves the alignment quality between vision and non-English features on various benchmarks.

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
