# OpenReview forum: "Multimodal LLM Enhanced Cross-lingual Cross-modal Retrieval"
_acmmm.org/ACMMM/2024/Conference — MM2024 Poster_

### Official Review · Reviewer_wr2a · 2024-05-02

**Rating:** 6
**Confidence:** 2

**Summary:**

This paper presents a novel Cross-lingual cross-modal retrieval model that leverages a multimodal large language model that generates detailed visual descriptions to use them as internal representations and  enhance visual representations and also utilizes English features to guide alignment between visual and the non-English modalities. This allows to tackle the gap between vision and text and especially the lower quality of non-English representations. Their extentive experiments demonstrate that their proposed method LECCR (LLM Enhanced Cross-lingual Cross-modal Retreival) method significantly improves the quality of the alignment between the visual features and the non-english textual features on a wide variety of various benchmarks that the authors evaluated their method on.

**Strengths:**

The idea is clever, the architecture and the methods are well described.
Results are consistent over multiple datasets and multiple languages and the ablation studies are provided to verify the effectiveness of the various components.
Personally I think that these kind of studies should always be emphasised given that methods on other languages always lack behind methods presented for the English language. Additionally, Cross-lingual models that can exploit the performance on the english language are always a nice addition to methods that are trained on datasets solely in another language. This is especially true for languages where the datasets typically lag behind those for the most popular languages.
I think that the evaluations are thorough and that the various graphs and even the attention map visualisation are nice addition and show how nicely each semantif slot nicely distinctly focus on ocal semantics within the image.

**Limitations:**

I personally don't see any significant limitations to this work. It appears to be sufficiently novel and of good quality for ICMR 2024 and it's definitely a good fit for such a conference where these methods that are Multimodal or crossmodal are at core of the interest.
To me the evaluation also appears sufficient but I might not be aware of the latests datasets in mulitmodal retrieval.

**Suitability:**

2

---

### Official Review · Reviewer_qJyK · 2024-05-24

**Rating:** 4
**Confidence:** 3

**Summary:**

The paper introduces a new method for cross-lingual cross-modal retrieval called LECCR, which enhances the alignment of visual and non-English textual representations using a Multimodal Large Language Model (MLLM). Without relying on manually labeled cross-modal data pairs, LECCR generates detailed visual content descriptions using the MLLM and aggregates them into multi-view semantic slots. These slots interact with visual features to enhance semantic information and narrow the semantic gap between modalities. Additionally, LECCR introduces soft matching under the guidance of English, establishing more comprehensive and reliable cross-modal correspondences. Experimental results demonstrate the effectiveness of LECCR on cross-lingual image-text and video-text retrieval benchmarks, showcasing its potential to improve cross-lingual cross-modal retrieval tasks.

**Strengths:**

- The paper introduces a dual-stream solution for cross-lingual cross-modal retrieval that utilizes a Multimodal Large Language Model (MLLM) to generate detailed visual content descriptions, thereby enhancing the semantic information within visual features, which is sufficiently novel.

-  Established models are employed for the multi-view visual-semantic interaction module, and the use of MLLM improves cross-modal alignment. The proposed concept of multi-view semantic slots adds depth to the model.
- The method has a broad range of applications, suitable for multilingual content retrieval, cross-cultural content analysis, and more.
- Comprehensive experiments are conducted on multiple benchmarks, including image-text and video-text retrieval tasks, to validate the method's effectiveness, universality, and robustness. Comparisons with existing methods and ablation studies further confirm the superiority of the method and the rationality of its design choices.

**Limitations:**

The proposed method may depend on the quality of MLLM generated descriptions, which may not always be accurate and affect the universality of the results.

**Suitability:**

3

---

### Official Review · Reviewer_H7Yq · 2024-05-24

**Rating:** 3
**Confidence:** 3

**Summary:**

This paper proposes LECCR, a novel solution for the CCR task. It aims to bridge the gap between modalities by incorporating the multi-modal large language model (MLLM) to generate detailed visual descriptions. Additionally, considering the lower quality of non-English representations, this paper utilizes English representations as guidance to improve the alignment between visual and non-English features by providing comprehensive inter-modal relationships.

**Strengths:**

1. Employing MLLM to improve the alignment between visual and non-English features is interesting.
2. These paper conducts extensive experiments on four image-text and video-text cross-modal retrieval benchmarks across different languages, demonstrating the effectiveness of method.

**Limitations:**

1. Model training is complex, and seven different objective functions were used during the training process. This could make the paper difficult to reproduce. Additionally, there is a lack of comprehensive ablation experiments and analysis, making it unclear what each loss function contributes to the model.

2. Utilizing large models to solve visual language tasks is a popular practice. For example [38] employs LLMs to generate auxiliary captions to enhance Text-Video Retrieval. The authors should compare and analyze their method against existing retrieval methods based on large models.

3. In the English Guidance module, the similarities between the source text and the vision are used at both local (semantic slots) and global (visual features) levels to guide the similarity of the target language text with the global vision (visual features). This setup is confusing. Why not constrain the similarity of the target text with the local vision (the semantic slots) as well? Furthermore, a more natural approach would be to use the global alignment of the source text with the visual features to constrain the global alignment of the target text with the visual features, and the local alignment of the source text with the semantic slots to constrain the local alignment of the target text with the semantic slots.

4.In Table 1, compared to CCLM [47], why the proposed model performs much worse on Multi30K but performs well on MSCOCO?

5.Is using MLLM to generate descriptions for all images offline or online?  This takes extra time. Compared with other dual-stream methods, how is the training speed and inference speed of this model?

6.How are the hyperparameters $\mu_1$ and $\mu_2$ determined?

**Suitability:**

3

---

### Official Review · Reviewer_k9PM · 2024-05-26

**Rating:** 3
**Confidence:** 4

**Summary:**

The strengths of the paper include understanding alignment among non-English and visual features by identifying a modality gap in cross-lingual cross-modal retrieval scenarios, proposing the LECCR framework for generating visual descriptions, conducting baseline evaluations using a set of pre-trained large multimodal models and showing how inter-modal relationships among non-English and visual features shape alignment via softened matching.

**Strengths:**

- The idea of multi-view visual-semantic interaction is promising for characterizing the alignment among non-English and visual features. And, so the cross-lingual cross-modal alignment via multi-level matching to consolidate both local and global features.
- Careful utilization of the single-stream and two-stream methods, and so the softened matching is an interesting way to guide the alignment among visual and non-English features.

**Limitations:**

- In Figure 1(b), the text feature in EN for guidance is a highly abstract caption, which reflects that the visual element slots attend to capture distinctive semantics, which wholly carries for retrieving related. Also, given the KL-divergence formulation in Eq. 21 and Eq. 22, it is plausible for one language to cater supervision but it doesn’t satisfy the invariance measure if more than one non-English language is used. The basic assumption should lie in the fact that it approximates the semantic distribution of each language which is not the case here. For instance, the textual captions are less semantic (such as the one in Figure 1(b) for EN/non-EN description which misses the information about a glass of beer).
- In the formulated similarity function in Eq. (8), authors should justify whether their multi-view regularization loss supports multiple instance learning [1] which is utilized in a cross-modal retrieval case [2]. Though, [1] lacks sparse supervision which is resolved in [3] by matching probability as a similarity function, however, Eq. (8) doesn’t reflect denser supervision scenarios. Authors should justify on the selection of their objective function and its aspects with the existing literature as aforementioned.

Comments:
- In L692, a missing information on inference time in this statement “our method is xx faster than CCLM in inference times”.


[1] Dietterich, T. G., Lathrop, R. H., & Lozano-Pérez, T. (1997). Solving the multiple instance problem with axis-parallel rectangles. Artificial intelligence, 89(1-2), 31-71.

[2] Song, Y., & Soleymani, M. (2019). Polysemous visual-semantic embedding for cross-modal retrieval. In Proceedings of the IEEE/CVF Conference on Computer Vision and Pattern Recognition (pp. 1979-1988).

[3] Chun, S., Oh, S. J., De Rezende, R. S., Kalantidis, Y., & Larlus, D. (2021). Probabilistic embeddings for cross-modal retrieval. In Proceedings of the IEEE/CVF Conference on Computer Vision and Pattern Recognition (pp. 8415-8424).

**Suitability:**

3

---

### Meta-Review · Area_Chair_LQeR · 2024-07-11

**Recommendation:** Accept (Poster)
**Confidence:** 5

**Metareview:**

**Conclusion: Accept as a Poster Paper**

After a thorough evaluation of the four independent reviews, I recommend accepting this sumission as a Poster Paper for ACM MM 2024. The paper proposes a novel LECCR framework for cross-lingual cross-modal retrieval, which effectively addresses the modality gap between non-English and visual features using a multimodal large language model (MLLM). This framework's ability to enhance semantic information and improve alignment between modalities across different languages makes it a valuable contribution to the field.

**Strengths:**

1. **Innovative Approach**: The paper introduces a dual-stream solution leveraging MLLM to generate detailed visual descriptions, enhancing the semantic information within visual features. This innovative approach contributes significantly to multimedia/multimodal processing.

2. **Comprehensive Experiments**: Extensive experiments across multiple benchmarks, including image-text and video-text retrieval tasks, validate the method's effectiveness, universality, and robustness. The comparisons with existing methods and ablation studies further confirm the method's superiority.

3. **Detailed Analysis and Visualizations**: The paper includes thorough evaluations and visualizations, such as attention maps, which illustrate how the proposed method focuses on local semantics within images. This adds depth to the understanding and validation of the model.

**Weaknesses:**

1. **Complexity and Reproducibility**: The model training process involves seven different objective functions, making it complex and potentially difficult to reproduce. There is also a lack of comprehensive ablation experiments to clarify the contribution of each loss function.

2. **Evaluation Consistency**: The method's performance on certain benchmarks, such as Multi30K, is inconsistent compared to others like MSCOCO. This raises questions about the robustness and generalizability of the approach across different datasets.

3. **Dependency on MLLM Quality**: The effectiveness of the proposed method depends on the quality of MLLM-generated descriptions, which may not always be accurate. This could affect the universality and reliability of the results.

In summary, despite the identified weaknesses, the strengths of the paper, particularly its innovative approach and comprehensive evaluation, justify its acceptance as a Poster Paper. The proposed LECCR framework offers a promising solution to cross-lingual cross-modal retrieval, making a notable contribution to the multimedia research community. Further refinement and additional evaluations could enhance the paper's impact and reproducibility, but its current state is sufficiently strong for presentation at ACM MM 2024.